# CHORDMIXER: A SCALABLE NEURAL ATTENTION MODEL FOR SEQUENCES WITH DIFFERENT LENGTHS

**Ruslan Khalitov, Tong Yu, Lei Cheng & Zhirong Yang**
Department of Computer Science
Norwegian University of Science and Technology
`{ruslan.khalitov,tong.yu,lei.cheng,zhirong.yang}@ntnu.no`

## ABSTRACT

Sequential data naturally have different lengths in many domains, with some very long sequences. As an important modeling tool, neural attention should capture long-range interaction in such sequences. However, most existing neural attention models admit only short sequences, or they have to employ chunking or padding to enforce a constant input length. Here we propose a simple neural network building block called ChordMixer which can model the attention for long sequences with variable lengths. Each ChordMixer block consists of a position-wise rotation layer without learnable parameters and an element-wise MLP layer. Repeatedly applying such blocks forms an effective network backbone that mixes the input signals towards the learning targets. We have tested ChordMixer on the synthetic adding problem, long document classification, and DNA sequence-based taxonomy classification. The experiment results show that our method substantially outperforms other neural attention models.[1]

## 1 INTRODUCTION

Sequential data appear widely in data science. In many domains, the sequences have a diverse distribution of lengths. For example, text information can be as short as an SMS limited to 160 characters or as long as a novel with over 500,000 words[2]. In biology, the median human gene length is about 24,000 base pairs (Fuchs et al., 2014), while the shortest is 76 (Sharp et al., 1985) and the longest is at least 2,300,000 (Tennyson et al., 1995). Meanwhile, long-range interactions between DNA elements are common and can be up to 20,000 bases away (Gasperini et al., 2020). Modeling interactions in such sequences is a fundamental problem in machine learning and brings great challenges to attention approaches based on deep neural networks.

Most existing neural attention methods cannot handle long sequences with different lengths. For efficient batch processing, architectures such as Transformer and its variants have been proposed, they usually assume constant input length. Otherwise, they have to use chunking, resampling, or padding to enforce the same input length. However, these enforcing approaches either lose much information or cause substantial waste in storage and computation. Even though some architectures, such as scaled dot-product (Vaswani et al., 2017), can deal with short sequences with variable lengths, they are not scalable to very long sequences.

In this paper, we propose a novel neural attention model called ChordMixer to overcome the above drawbacks. The new neural network takes a sequence of any length as input and outputs a tensor of the same size. Moreover, ChordMixer is scalable to very long sequences (we demonstrate lengths up to 1.5M in our experiments).

ChordMixer comprises several modules or blocks with a simple and identical architecture. Each block has a Multi-Layer Perceptron (MLP) layer over the sequence channels and a multi-scale rotation layer over the element positions. The rotation has no learnable parameters, and therefore the model size of each block is independent of sequence length. After $\log_2 N$ ChordMixer blocks, every number in the output has a full receptive field of all input numbers for length $N$.

---

[1] Code is publicly available at https://github.com/RuslanKhalitov/ChordMixer
[2] https://en.wikipedia.org/wiki/List_of_longest_novels

We compared ChordMixer with Transformer and many of its variants in three tasks over sequential data: synthetic adding problem, long document classification, and DNA sequence-based taxonomy classification. Our method wins in nearly all tasks, which indicates that ChordMixer mixes well the signals in long sequences with variable lengths and can serve as a transformation backbone in place of conventional neural attention models.

The next section will briefly review the definitions and related work. We present the ChordMixer, including its design, properties, and implementation details, in Section 3. Settings and results of three groups of experiments are provided in Section 4. In Section 5, we conclude the paper and discuss future work.

## 2 BACKGROUND AND RELATED WORK

A sequential data instance, or a sequence, is a one-dimensional array of sequence elements or tokens. In this work, tokens are represented by vectors of the same dimensionality. Therefore a sequence can be treated as a matrix $x \in \mathbb{R}^{d \times N}$, where $N$ is the sequence length and $d$ is the token dimensionality (or the number of channels). Each sequence may have a different length in a data set, and the distribution of $N$ can have a high range.

Neural attention or mixing is a basic function of a neural network, which transforms a data tensor into another tensor toward the learning targets. In the self-attention setting, the input and output tensors usually have the same size. Without losing information, neural attention should have a full receptive field; that is, each output number can receive information from all input numbers. However, naively connecting each pair of input and output numbers is infeasible. Self-attention of a sequence $x$ with the naive implementation requires $N^2 d^2$ connections, which is too expensive when $Nd$ is large.

Most existing neural attention methods employ two-stage mixing to relieve the expense due to the full connections. For example, the widely used Transformer model (Vaswani et al., 2017) alternates the token-wise and position-wise mixing steps. The connections are limited in each step, but the receptive field becomes full after one alternation. However, Transformer has a quadratic cost to sequence length because it fully connects every token pair.

Numerous approximation methods of Transformer have been proposed to reduce the quadratic cost. For example, Longformer (Beltagy et al., 2020) and ETC (Ainslie et al., 2020) use a learnable side memory module that can access multiple tokens at once; Nyströmformer (Xiong et al., 2021) uses a few landmarks as surrogates to the massive tokens; downsampling methods also include Perceiver (Jaegle et al., 2021) and Swin Transformer (Liu et al., 2021; 2022); Performer (Choromanski et al., 2020) and Random Feature Attention (Peng et al., 2021) approximate the softmax kernel by low-rank matrix products; Switch Transformer (Fedus et al., 2021) and Big Bird (Zaheer et al., 2020) use sparse dot-products at multiple layers. A more thorough survey can be found in Tay et al. (2022). However, the approximation methods still follow the scaled dot-product approach in the original Transformer, and thus their performance remains mediocre or inferior (Gu et al., 2022; Khalitov et al., 2022; Yu et al., 2022a).

## 3 CHORDMIXER

In this work, we go beyond the scaled dot-product approach and aim to develop a neural attention model with the following properties:

- *full-receptive field*: every output number is mixed directly or indirectly from all input numbers of a sequence;

- *scalability*: the new method can give accurate predictions for very long sequences;

- *decentrality*: the new design is decentralized, where no sequence element or position is more central or closer to the output;

- *length flexibility*: the model can handle sequences of diverse lengths without extra preprocessing such as chunking, resampling, or padding.

Moreover, we do not assume that useful patterns appear only locally, and we try to learn all possible interactions from data. Therefore we avoid operations such as pooling over local windows of the sequences.

## 3.1 INSPIRATION FROM P2P NETWORK

We find an analogous solution in the Peer-to-Peer (P2P) communication network to achieve the above properties (Stoica et al., 2001). Full-connection is a naive communication protocol when every peer tries to communicate with the other $N-1$ peers. However, the naive protocol leads to huge routing tables, and message flooding often causes wasteful communication.

As a solution, one of the best-known protocols in P2P is Chord, where all peers are organized in a circle. At each hop of lookup, the $i$-th peer will communicate with the neighbors $i + 2^0, \ldots, i + 2^m$ at multiple scales and with modulus $N$, where $m = \lceil \log_2 N \rceil$. See Figure 1 (a) for illustration. The information collected from the communication will be stored and communicated at the next hop. In the distributed setting, each peer is able to collect information (directly or indirectly) from all other peers after $O(\log_2 N)$ hops (Stoica et al., 2001).

## 3.2 CHORDMIXER BUILDING BLOCK AND NETWORK ARCHITECTURE

We can mimic the P2P Chord protocol with an artificial neural network by treating sequence tokens as peers, network blocks as hops, and transformations between blocks as communications. We called the resulting building block ChordMixer.

We divide the sequence rows or channels into $M = \lceil \log_2 N \rceil + 1$ tracks, where each track has about $d/M$ channels. Then each ChordMixer block contains two layers for an input sequence $x^{\text{in}} \in \mathbb{R}^{d \times N}$:

1. *(Rotate)*: the layer rotates the tracks at different scales ($z \in \mathbb{R}^{d \times N}$):

$$z_{ij} = x_{ik}^{\text{in}} \tag{1}$$

   for $i = 1, \ldots, d$ and $j = 1, \ldots, N$, where

$$k = \begin{cases} j & \text{if } t_i = 1 \\ \left(j + 2^{t_i - 2}\right) \mod N & \text{if } t_i > 1 \end{cases} \tag{2}$$

   with $t_i$ the track index of the $i$-th channel (starting from 1). Here we add the self-loop (a non-rotated track) to the Chord protocol to include the same token in mixing. The Rotate forward pass is illustrated in Figure 1 (b)-(d).

   The Rotate layer is cheap to implement. The copy operations from $x$ to $z$ can run in parallel over $i, j, k$, and thus can efficiently be implemented by e.g., CUDA. It can also be done in PyTorch with the `roll` function. The backward pass is simply a reverse rotation: $\partial \mathcal{J} / \partial x_{ik}^{\text{in}} = \partial \mathcal{J} / \partial z_{ij}$ for a learning objective $\mathcal{J}$.

2. *(Mix)*: transforms every token (column) with a neural network $f$

$$x_{:j}^{\text{out}} = f(z_{:j}; w) \tag{3}$$

   for $j = 1, \ldots, N$, where $w$ is the network weights of $f$. In the experiments, we simply used a two-layer MLP to implement $f$ with a hidden dimension $h$ and a GELU nonlinearity. We also apply dropout between the Rotate and Mix steps.

A ChordMixer network consists of $\lceil \log_2 N_{\max} \rceil$ ChordMixer blocks, where $N_{\max}$ is the length of the longest sequence or the longest range of possible interactions. Each block has an identical architecture (with $\lceil \log_2 N_{\max} \rceil + 1$ tracks) but a different $f$ network. Residual connection is applied around each ChordMixer block.

A ChordMixer network forms a backbone of a neural learner. We believe after the mixing, elements at every position summarize good information from all input positions. We can then use a lightweight predictor (e.g., classifier or regressor) on the individual output tokens and integrate their predictions by ensemble learning. In this work, we use linear predictors and ensemble averaging. Because the linear operation and averaging are exchangeable, we can first take the mean of output tokens and then apply the linear predictor.

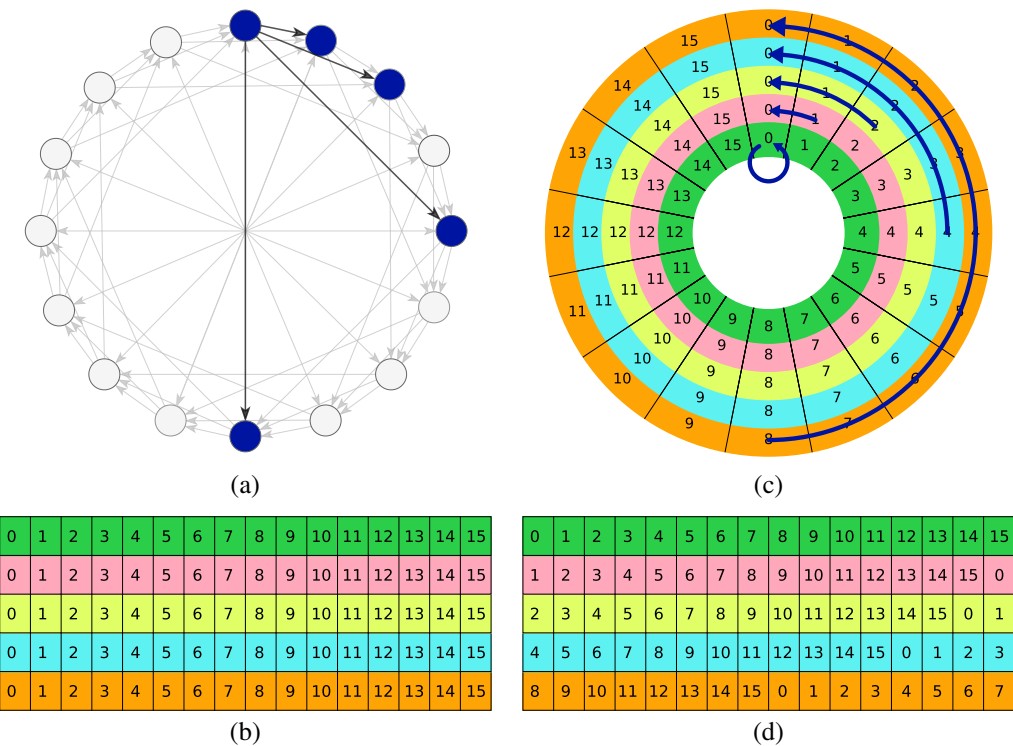

Figure 1: Illustration of the Rotate step in ChordMixer for $N = 16$: a) the original Chord protocol, where blue dots are peers, and arrows indicate communications at each hop, b) the sequence before rotation, where the numbers indicate the original position in each track, c) the rotations in different bands, where the arcs indicate the rotation sources of new Position 0, and d) the rotated tracks. Afterward, each column will be fed to the Mix step of ChordMixer.

We arrange batch learning and inference where the sequences in a batch have the same $\lceil \log_2 N \rceil$ value such that they pass through the same number of blocks. We rotate sequences in a batch in each block separately, then concatenate the rotated for MLP mixing only once. Finally, the mixed tensor is de-concatenated to sequences for block output. There is no padding or chunking in the procedure. Here the key point is that the Rotate layer is cheap, while the Mix layer benefits from the batch-mode acceleration. The same trick does not apply to Transformer-like approaches where dot products and other attention types are expensive.

### 3.3 ANALYSIS

**Proposition 3.1.** *The receptive field becomes full for every output number after $O(\log_2 N)$ blocks.*

The proposition is a straightforward corollary of Theorem 2 in the Chord paper (Stoica et al., 2001), and we skip the proof here. By this result, not all blocks are needed for every sequence. A length-$N$ sequence passes through only the first $\lceil \log_2 N \rceil$ blocks, and therethrough the gradient back-propagation.

**Definition 3.2.** *A neural network for $D$-dimensional inputs is full-rank if the network comprises linear layers and elementwise nonlinearities, and all the linear layers have at least rank $D$.*

**Proposition 3.3.** *A ChordMixer block is full-rank on flattened $x^{in}$ if $f$ is full-rank.*

The proof is given in Appendix A. By this result, our method differs from typical bottleneck designs in conventional neural networks that contain low-rank projections. Instead, ChordMixer is a wide neural network that directs the input signal towards the learning targets in a broad way.

Next, we analyze the complexity of ChordMixer:

- Because the Rotate layer is parameter-free, all learnable parameters in a ChordMixer block are $w$. For the two-layer MLP implementation of $f$, there are $O(dh)$ parameters in one block. Overall, a ChordMixer network has $O(dh \log_2 N_{\max})$ learnable parameters.

- The Mix step dominates the running time. The MLP mixing in each block costs $O(dhN)$ and thus $O(dhN \log_2 N)$ for all blocks.

- The memory cost for a length-$N$ sequence passing a ChordMixer network is $O(dN \log_2 N)$. If we employ the Reversible Residual Network trick (Gomez et al., 2017) to avoid storing intermediate activations, the memory cost can be reduced to $O(dN)$.

### 3.4 Comparison to related work

A recent neural attention model called Paramixer (Yu et al., 2022a) also uses the Chord protocol for token mixing. It employs similar mixing MLPs over channels. Differently, Paramixer decomposes an attention matrix into several sparse factors. The non-zero structure of the sparse factors is specified by the Chord protocol, and their values are parameterized by multilayer perceptrons. Paramixer assumes that the input lengths are the same. Otherwise, truncation or padding is required to ensure the same length.

PoolFormer (Yu et al., 2022b) is another neural attention model which employs a non-parametric operator over positions. Different from the track rotations in ChordMixer, PoolFormer applies pooling over a local window around every position. The pooling connects all elements within the window and thus has a similar role for information propagation. However, defining such windows requires extra assumptions on locality. Despite success in vision and speech, the locality assumptions may not apply to other types of sequential data, as we shall see in the experiments.

Some other methods organize the sparse connections in a hierarchical manner (e.g., Goller & Küchler, 1996; Choi et al., 2018; Lea et al., 2016). Although they can achieve a full-receptive field with even fewer connections, the hierarchy breaks the symmetry: some elements (or positions) become closer to the output than others. The artifact is undesired for neural attention because a good attention model should allow each sequence element to have a (nearly) equal chance to interact with the others. In contrast, the design of ChordMixer is *decentralized*, where no element or position is below or above another. When passing through a ChordMixer network, every output position can receive information from all input positions with nearly the same probability (see Appendix G).

There is a theoretical equivalence between ChordMixer and a comprehensive convolution: each output track equals the sum of $\log_2 N$ convolutions, where each convolution applies to the corresponding input track. The first of the $\log_2 N$ convolution is $1 \times 1$, while each of the rest uses a different dilation, kernel size 1, circular padding, and causal connection (self-excluding). However, such a convolution is impractical because 1) no current software supports such a complicated convolution, and 2) the convolutions have to be run track by track, which is much slower than the proposed rotate-mix steps.

## 4 Experiments

In this section, we test the scalability and performance of ChordMixer in three different domains: 1) an arithmetic task with long-range dependence between sequence elements, 2) long text document classification, and 3) taxonomy classification based on DNA sequences. In the tasks, ChordMixer is compared with many other existing neural attention approaches, including Transformer (Vaswani et al., 2017), Reformer (Kitaev et al., 2020), Linformer (Wang et al., 2020), Nyströmformer (Xiong et al., 2021), Luna (Ma et al., 2021), Longformer (Beltagy et al., 2020), Cosformer (Qin et al., 2022), Poolformer (Yu et al., 2022b), and Structured State-Space sequence model (S4) (Gu et al., 2022). For all methods, we used a validation set to tune their hyperparameters (details in Appendix C). For completeness, we also include Recurrent Neural Network (LSTM; Hochreiter & Schmidhuber, 1997), Recursive Neural Network (TreeLSTM; Choi et al., 2018), conventional convolution network (1D CNN), hierarchical convolutional network (TCN; Lea et al., 2016) in the comparison. All experiments were conducted on a Linux machine with 8xNVIDIA Tesla V100-32GB GPUs.

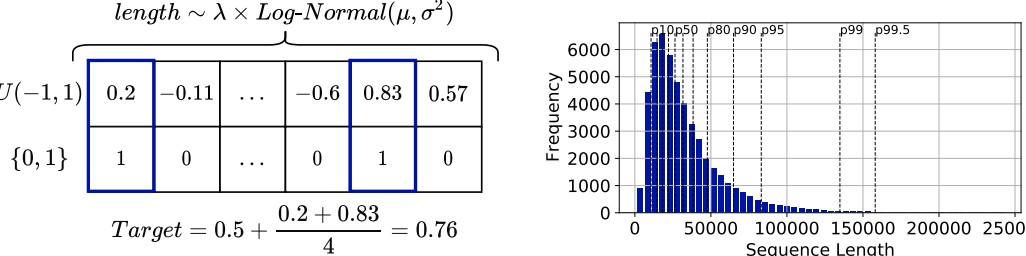

Figure 2: (Left) one instance of the Adding problem with variable lengths. Lengths are sampled from a Log-Normal distribution and times with a base length ($\lambda$). (Right) example of sequence lengths distribution for $\lambda = 16k$. The dashed vertical lines correspond to the percentiles of the distribution.

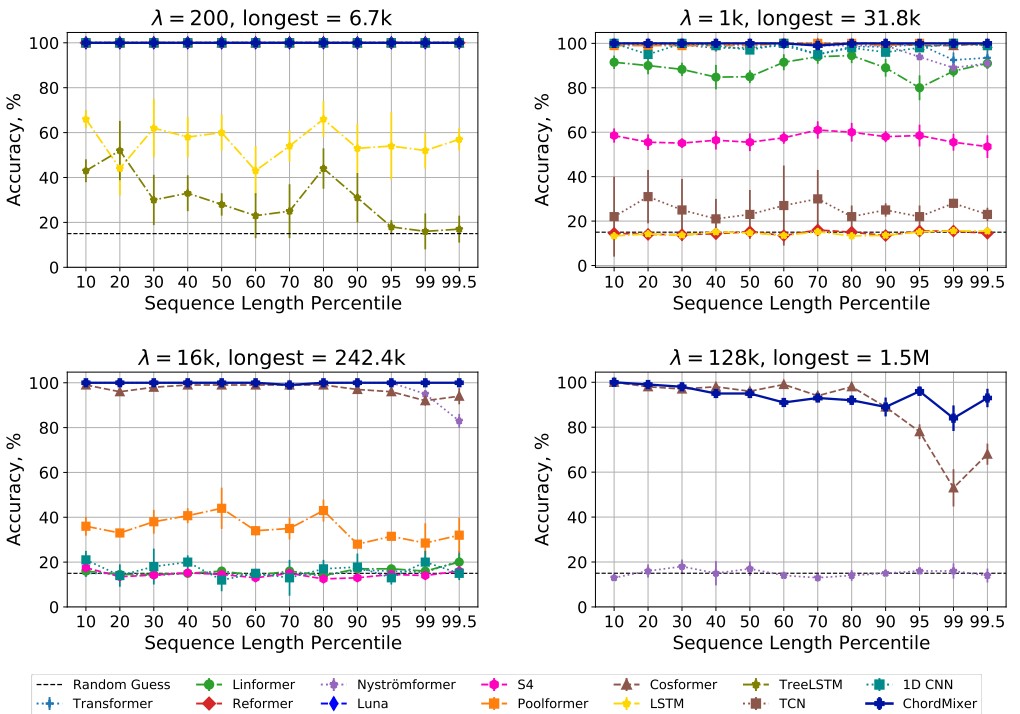

Figure 3: Prediction accuracies of the Adding problem with sequences of variable size. $\lambda$ controls the mean length of the sequences. The mean scores and error bars across 3 runs are plotted. For $\lambda = 16k$ and $\lambda = 128k$, We drop some architectures in the plots because they ran out of memory.

## 4.1 ADDING PROBLEM WITH VARIABLE LENGTHS

The Adding problem is a synthetic task inspired by Hochreiter & Schmidhuber (1997). This is a regression task used to test networks ability to detect relevant tokens and perform the sum operation on them. Each element of an input sequence is a pair of numbers $(a_i, b_i)$, where $a_i \sim U(-1, 1)$, $b_i \in \{0, 1\}$, $i = 1, \ldots, N$. We generated signals at two randomly selected positions $t_1$ and $t_2$ such that $b_{t_1} = b_{t_2} = 1$ and $b_i = 0$ elsewhere. The learning target is $y = 0.5 + \dfrac{a_{t_1} + a_{t_2}}{4}$. Unlike Hochreiter & Schmidhuber (1997); Khalitov et al. (2022), we generated sequences with different length $N = \text{round}(\lambda \cdot \zeta)$ and $\zeta \sim \text{LogNormal}(\mu, \sigma^2)$, where $\mu = 0.5$, $\sigma = 0.7$, and $\lambda$ is a base length. The resulting length distribution is left-skewed and has a wide spread (see Figure 2 right). The

variable lengths bring extra challenge to the task. In evaluation, a network prediction $\hat{y}$ is considered correct if $|y - \hat{y}| < 0.04$. A sequence example is illustrated on Figure 2 (left).

In the experiment setup, we used four different $\lambda$'s: 200, 1k, 16k, and 128k. For each $\lambda$, we generated 60,000 learning instances (sequences and their targets). A larger $\lambda$ corresponds to a more difficult task. For example, when $\lambda = 200$, the longest sequence has a length of 6.7k, while $\lambda = 128k$ leads to the longest sequence up to 1.5 million elements. Figure 2 (right) shows the histogram of sequence lengths with $\lambda = 16$k.

Figure 3 shows the comparison results. The tested models worked perfectly in the easiest setup ($\lambda = 200$). With increasing $\lambda$'s, some methods started to fail. For example, Reformer performed as poorly as random guessing for $\lambda = 1000$. Transformer, Nyströmformer, Linformer, and S4 also had more or less degraded accuracies when $\lambda = 1000$. When $\lambda = 16k$ and the longest sequence becomes 242K, three methods (Transformer, Reformer, and Luna) ran out of memory and could not finish training. Only ChordMixer, Cosformer, and Nyströmformer achieved more than 80% accuracy, while the others got close to random guess. ChordMixer was the only method that reached nearly 100% for $\lambda = 16k$. When $\lambda = 128k$, only ChordMixer and Cosformer finished training, while the methods had out-of-memory errors. ChordMixer still achieved almost 90% accuracy for all sequences, while Cosformer became inaccurate for the 10% longest sequences.

## 4.2 Long document classification

This task measures the performance of Chord-Mixer in modeling complex long-term dependencies for very long texts. The data set consists of a collection of academic papers and is publicly available on Github. Following He et al. (2019), we used four classes of the documents: cs.AI, cs.NE, math.AC, math.GR, which results in a data set of 11956 documents. We split the data set into 70% training, 20% validation, and 10% test sets. Unlike the original research (He et al., 2019), we encoded each document on the character level and formed a more challenging task. The minimum, median, and maximum lengths of the documents are 4.5k, 39k, and 131k, respectively. All compared methods except ChordMixer used zero padding to align with the maximum length.

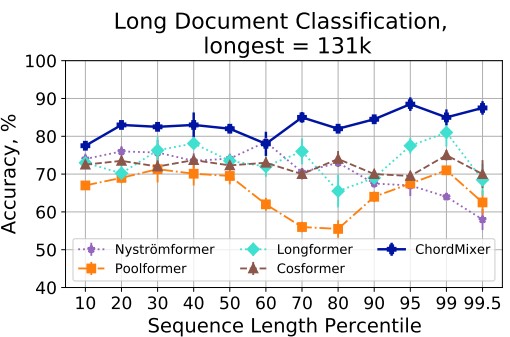

Figure 4: Accuracies in the Long Document Classification task at different length percentiles.

Figure 4 shows the classification accuracies. ChordMixer performs the best, where it is the only method that achieves nearly or above 80% accuracy at all percentiles. Nyströmformer is as good as ChordMixer at 60% sequence length percentile, while it becomes worse and worse for longer documents. Cosformer achieves around 70% accuracy at all percentiles. The accuracies of PoolFormer are lower than 70% almost everywhere and as low as 55%. The accuracies of Longformer fluctuate around 70%. Five other compared methods (Transformer, Linformer, Reformer, S4, and Luna) ran out of memory, and we have to drop them from the comparison figure.

## 4.3 DNA sequence-based taxonomy classification

Next, we test ChordMixer and other compared methods in biology. We have downloaded DNA sequences from the Genbank database[3] as well as their taxonomic labels. Then we organized four binary classification tasks (ordered by the maximum sequence length):

1. **Carassius vs. Labeo.** This is a small data set with 7263 genes for Carassius (a kind of fish) and 6718 sequences for Labeo (another kind of fish). The shortest sequence has 79 base pairs, while the longest one has 100101. In terms of the required scalability of the tested models, this task is the easiest problem within this section.

---

[3] https://www.ncbi.nlm.nih.gov/genbank/

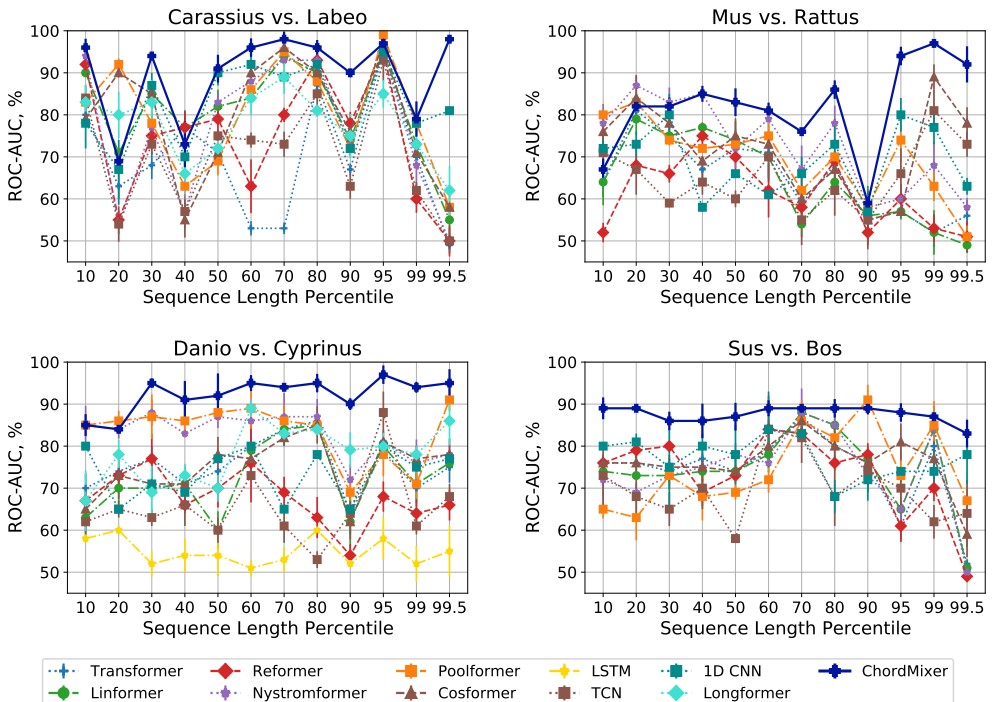

Figure 5: Test ROC-AUCs on the DNA-based taxonomy classification tasks. The score is the average of three runs with different random seeds. Plots are better read in colors.

2. **Mus vs. Rattus.** This data set is the heaviest among used within this problem setup at the genus level. The classes are strongly imbalanced, differing in size by almost a factor of two. The task is to classify a gene as a rat (rattus) or a mouse (mus). There are 275636 Mus and 133310 Rattus sequences, with minimum and maximum sequence lengths of 32 and 261093, respectively.

3. **Danio vs. Cyprinus.** Both genera belong to the cyprinidae family, where danio is small freshwater fish, and cyprinus is a genus of carps. The shortest gene consists of 34 bases, while the longest has 261943. Classes have imbalanced: 47135 Danio and 83321 Cyprinus sequences.

4. **Sus vs. Bos.** Both genera belong to the Vertebrate organisms group, where sus and bos roughly correspond to domestic pigs and cattle, respectively. Although the genera are nearly balanced (51973 sequences for bos and 50027 for sus), the data set is the most challenging because the lengths are highly different (shortest 63 and longest 447010).

We chose these four classification tasks because the taxa have a similar length distribution, and it is difficult to classify them only by sequence length (statistics in Appendix B).

We have used Area under the receiver operator curve (ROC-AUC) as the evaluation metric to overcome the class imbalance effect. All models were trained using cross-entropy loss, with empirical class weights calculated on the training set. The train/validation/test split was stratified to ensure equal class distribution within these data sets. We have applied zero padding to the maximum sequence length for all compared models except ChordMixer. For Transformer, Reformer, and Linformer, the sequences were truncated to the length such that they do not throw an out-of-memory error.

Figure 5 shows the ROC-AUC results of the four taxonomy classification tasks. ChordMixer stands at the top or near the top at almost all sequence length percentiles for all data sets. Especially at the right end of every plot, our method achieves substantially higher ROC-AUC than all other approaches, which shows that ChordMixer has a clear win over those methods which require zero padding. For the smallest data set (Carassius vs. Labeo), several methods are comparable to ChordMixer for short

sequences. However, starting from the 99 percentile (corresponds to sequence length 5630), all other compared methods experience a significant performance drop. For the longest part (99.5 percentile in Sus vs. Bos), ChordMixer achieves 89% ROC-AUC, while all other methods are below 70%. With increasing maximum sequence length, ChordMixer wins at more length percentiles. For Danio vs. Cyprinus and Sus vs. Bos, ChordMixer is clearly the best at nine out of ten listed percentiles. Interestingly, ChordMixer is the only method that achieves stable ROC-AUC (around 84-89%) across various lengths for the Sus vs. Bos data set.

## 4.4 ABLATION STUDY

In the above experiments, every learning system comprises a neural attention part and a predictor (regressor or classifier). Here we perform an ablation study to show that the main improvement comes from the ChordMixer as the neural attention part and not from the predictor.

We chose the Carassius vs. Labeo taxonomy classification task for the ablation study. Besides ChordMixer, we included two other compared methods, Poolformer and Cosformer, because their overall accuracy or ROC-AUC seem to be more competitive than the rest (others' results in Appendix E to avoid clutter). Each neural model was combined with two different predictors: AVG) averaging followed by a linear classifier as we used in the above ChordMixer results, and CLS) using a classification token in the neural attention and then applying a linear classifier. For Poolformer and Cosformer, we also tried a linear classifier on the flattened output from neural attention (FLAT).

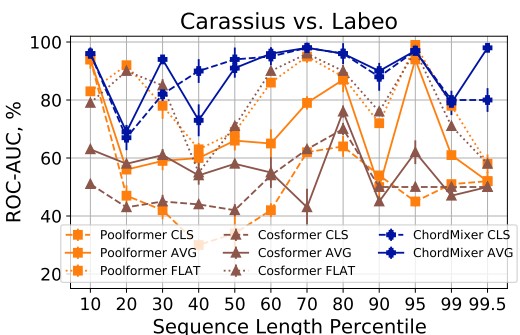

Figure 6: Ablation study on different combinations of neural attention models and classifiers.

In Figure 6, ChordMixer, either with CLS or AVG, wins for all sequence lengths except the 20 and 95 percentiles. Even with the same classifier, both Poolformer and Cosformer perform worse than ChordMixer. The FLAT strategy sometimes improves ROC-AUC for the competing models but still cannot surpass ChordMixer at most percentiles.

## 5 CONCLUSION AND FUTURE WORK

We have proposed an effective neural attention model ChordMixer for long sequences with variable lengths. The building block of our model contains two simple layers: non-parametric multi-scale rotation and channel mixing. ChordMixer does not require chunking or padding at all and easily scales up to a maximum sequence length of 1.5 million in our work. We have provided experiments from different domains, where our method has outperformed many other recent approaches.

In our method, sequences pass through a variable number of ChordMixer blocks, depending on their lengths. We have not found practical issues so far for such a non-conventional setting. However, it challenges theoretical analysis because the mapping function space becomes more complicated, which demands more advanced mathematical tools.

In our design, we do not assume any local patterns in the data to endow nearly equal chance for all tokens to interact with each other. If specific local patterns prevail in the data (e.g., Markovian property in images), our method might require more training data than approaches that use locality as an inductive bias. In the future, we could study how to learn such patterns using our model with massive self-supervised masked training. Relative position information could be incorporated (in an economical way) to facilitate pattern learning.

## 6 ETHICS STATEMENT

It is implausible that our work would bring potential negative societal impacts. We have proposed a basic methodology for machine learning. Our research has no connection to weapons, surveillance systems, adverse experiments, or privacy and security issues. Therefore, there is little risk of harmful applications for living beings and human rights. Moreover, our findings benefit the environment by substantially reducing the computational complexity in neural attention models.

Our work also fulfills the requirements from the ICLR Code of Ethics. The research does not use human-derived data. For the synthetic data set generated by us, we publish the data generation code (under the MIT license). For the public document corpus and genome (non-human) data set, we have followed the usage requirement from the data provider and cited their contributions. There are no domain-specific considerations when working with high-risk groups because our work does not involve research on human beings. Our data does not contain offensive content. Moreover, our work complies with GDPR because we do not reveal sensitive personal information.

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

## A    PROOF OF PROPOSITION 3.3

*Proof.* A ChordMixer block contains a Rotate layer and a Mix layer. Below we show that both layers are full-rank.

The Rotate layer re-orders the numbers in the matrix. Therefore, it is equivalent to left-multiplying the flattened $x^{\text{in}}$ with the corresponding permutation matrix. Because permutation is linear and full-rank, the Rotate layer is full-rank.

The Mix layer applies $f$ on the columns (tokens). By the assumption, $f$ is a full-rank network that contains element-wise non-linearities and linear operators. Each linear operator is equivalent to left-multiplying a block-wise matrix with the column-major flattened input $x$, where each block in the multiplying matrix is the linear weights $W$. The following illustrates the first linear operator, where its flattened output equals

$$
\begin{bmatrix}
W & 0 & 0 & \cdots & 0 \\
0 & W & 0 & \cdots & 0 \\
0 & 0 & W & \cdots & 0 \\
\vdots & \ddots & \ddots & \ddots & \vdots \\
0 & 0 & 0 & \cdots & W
\end{bmatrix}
\begin{bmatrix}
x_{:1} \\
x_{:2} \\
x_{:3} \\
\vdots \\
x_{:N}
\end{bmatrix}
\tag{4}
$$

Because $W$ has at least rank $d$, the whole multiplying matrix has at least rank $dN$. Therefore, the Mix layer is full rank. $\qquad\square$

## B    DATA PREPARATION

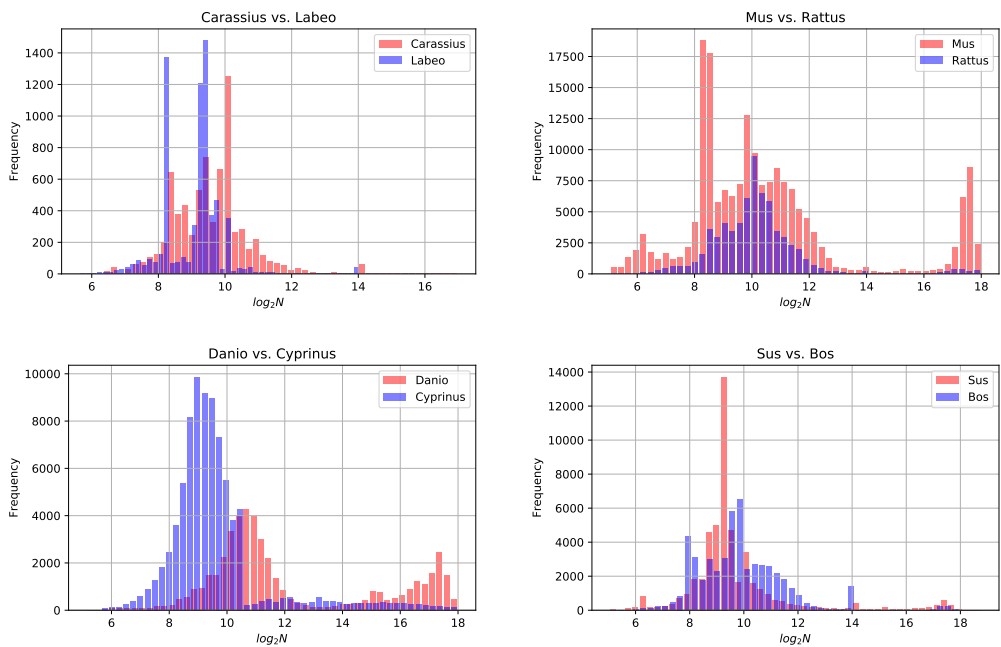

Figure 7: Sequence lengths distribution for the DNA-based taxonomy classification task.

Here we provide extra information about data preparation in the three groups of experiments

**Adding problem.** For each setup, we generated $60,000$ data instances (including training, validation, and test sets), except for the hardest task. The reason for the exception is that all sequences are padded to the maximum length, and storing so many long tensors requires enormous hard disk space and heavy RAM for reading. With $\lambda = 128$k, the longest among $60,000$ generated sequences has over three million elements, which can cause all compared methods except ChordMixer to run out of

memory. Therefore, we reduced the number of generated sequences to $12,000$ for $\lambda =$128k for a more reasonable comparison.

Because there are only two input channels in the raw sequences, we used a linear module to augment the embedding space to higher dimensions. For ChordMixer, the embedding layer outputs $d$ channels, where $d$ equals TrackSize times $\log_2 N_{\max}$. The number of channels for the other compared methods, ranging from 32 to 128, are tuned by hyperparameter search (see Section C for details).

**Long document classification.** There are a few documents are extremely long (longest=$2,483,331$). Because all compared methods except ChordMixer padded zeros to align with the longest, they are easily prone to an out-of-memory error. Therefore, we used the maximum length of 131k for a more reasonable comparison. There are 56 among 11956 documents ($\leq 0.47\%$) longer than 131k, and we discarded their tailing part beyond 131k (no compared method has access to the discarded tailing part).

**DNA-based taxonomy classification.** The GenBank collection contains massive genes from many taxa. However, not all branches in the taxonomy tree are suitable for comparing neural attention models. Quite often, the taxa can easily be separated by the gene lengths. Here we particularly chose four classification tasks where the involved taxa have overlapped length distributions. See Figure 7 for sequence length histograms in the tasks, where we can see that it is hard to classify the taxa using their lengths solely.

## C    MODEL CONFIGURATIONS AND TRAINING DETAILS

In this section, we provide the full set of hyperparameters used to train models and report their results. The final hyperparameters were chosen based on the Bayesian optimization algorithm with a 32GB GPU memory constraint. After the best hyperparameters selected by the minimum validation loss, we applied the corresponding model to the test set and report its performance.

The adding problem is a regression task, where we used the MSE loss. The other two tasks are classification, where we used the cross-entropy loss, with class weights, calculated on the training set, to overcome the class imbalance effect. Adam optimizer with task- and model-specific learning rates was used for all problems.

The training configuration and hyperparameters for each neural network and problem are summarized in Table 1 and Table 2. For the adding problem, we used the same hyperparameters across all setups.

Table 1: The final competitors' hyperparameters used in experiments. $L$, $H$, $E$, and $P$ refer to number of layers, number of heads, embedding size, pooling strategy, respectively. Dash ("-") indicates that the corresponding parameter is not present in the architecture.

| Task. | Model | $L$ | $H$ | $E$ | $P$ | LearningRate | BatchSize |
|---|---|---|---|---|---|---|---|
| Adding | Transformer | 2 | 4 | 64 | FLAT | 0.0005 | 5 |
|  | Linformer | 1 | 2 | 128 | FLAT | 0.0001 | 10 |
|  | Reformer | 2 | 2 | 64 | FLAT | 0.0003 | 5 |
|  | Cosformer | 2 | 2 | 128 | FLAT | 0.0005 | 5 |
|  | Poolformer | 2 | 2 | 128 | FLAT | 0.0005 | 10 |
|  | Luna | 4 | - | 32 | FLAT | 0.0005 | 5 |
|  | S4 | 4 | - | 32 | FLAT | 0.0005 | 5 |
|  | Nyströmformer | 2 | 2 | 128 | FLAT | 0.0002 | 5 |
| Long Document | Cosformer | 1 | 2 | 64 | FLAT | 0.0005 | 10 |
| Classification | Poolformer | 1 | 2 | 64 | FLAT | 0.0005 | 10 |
|  | Nyströmformer | 2 | 2 | 128 | FLAT | 0.0005 | 10 |
| Genbank | Transformer | 1 | 2 | 128 | TRUNC | 0.0005 | 4 |
| C/L | Linformer | 1 | 2 | 128 | TRUNC | 0.0005 | 4 |
|  | Reformer | 1 | 2 | 64 | TRUNC | 0.0005 | 4 |
|  | Cosformer | 2 | 2 | 128 | FLAT | 0.0003 | 4 |
|  | Poolformer | 2 | 2 | 128 | FLAT | 0.0003 | 4 |
|  | Nyströmformer | 2 | 2 | 128 | FLAT | 0.0002 | 4 |
| Genbank | Transformer | 1 | 2 | 128 | TRUNC | 0.0005 | 4 |
| M/R | Linformer | 1 | 2 | 128 | TRUNC | 0.0005 | 4 |
|  | Reformer | 1 | 2 | 64 | TRUNC | 0.0005 | 4 |
|  | Cosformer | 2 | 2 | 128 | FLAT | 0.0005 | 4 |
|  | Poolformer | 2 | 2 | 128 | FLAT | 0.0005 | 4 |
|  | Nyströmformer | 2 | 2 | 128 | FLAT | 0.0005 | 4 |
| Genbank | Transformer | 1 | 2 | 128 | TRUNC | 0.0005 | 4 |
| D/C | Linformer | 1 | 2 | 128 | TRUNC | 0.0005 | 4 |
|  | Reformer | 1 | 2 | 64 | TRUNC | 0.0005 | 4 |
|  | Cosformer | 2 | 2 | 64 | FLAT | 0.0003 | 4 |
|  | Poolformer | 2 | 2 | 64 | FLAT | 0.0003 | 4 |
|  | Nyströmformer | 2 | 2 | 64 | FLAT | 0.0002 | 4 |
| Genbank | Transformer | 1 | 2 | 128 | TRUNC | 0.0005 | 4 |
| S/B | Linformer | 1 | 2 | 128 | TRUNC | 0.0005 | 4 |
|  | Reformer | 1 | 2 | 64 | TRUNC | 0.0005 | 4 |
|  | Cosformer | 2 | 2 | 64 | FLAT | 0.0003 | 4 |
|  | Poolformer | 2 | 2 | 64 | FLAT | 0.0003 | 4 |
|  | Nyströmformer | 2 | 2 | 64 | FLAT | 0.0002 | 4 |

Table 2: Hyperparameters used to train ChordMixer. $h$ and $P$ refer to hidden layer size and pooling strategy, respectively. The embedding size is TrackSize$\times \log_2 N_{\max}$.

| Task. | $h$ | TrackSize | $P$ | LearingRate | BatchSize |
|---|---|---|---|---|---|
| Adding | 128 | 16 | AVG | 0.0001 | 2 |
| Long Document Classification | 196 | 16 | AVG | 0.0002 | 2 |
| Genbank C/L | 128 | 16 | AVG | 0.0001 | 2 |
| Genbank M/R | 128 | 16 | AVG | 0.0001 | 2 |
| Genbank D/C | 128 | 16 | AVG | 0.0001 | 2 |
| Genbank S/B | 128 | 16 | AVG | 0.0001 | 2 |
| LRA Image | 420 | 28 | AVG | 0.01 | 100 |
| LRA Text | 96 | 16 | AVG | 0.007 | 90 |
| LRA Listops | 88 | 8 | AVG | 0.007 | 150 |
| LRA Retrieval | 128 | 12 | AVG | 0.007 | 90 |
| LRA Pathfinder | 320 | 24 | AVG | 0.005 | 150 |
| LRA PathfinderX | 128 | 10 | AVG | 0.001 | 120 |

## D    CHORDMIXER ATTENTION EXAMPLE

Every ChordMixer block outputs a $d \times N$ matrix. We can visualize the matrices to study how the model attends to different positions. We trained a neural network for the Adding problem with $\lambda = 200$, where the shortest and longest sequences have lengths 32 and 6655, respectively. Figure 8 shows source sequences and the resulting matrices after applying the first and the last ChordMixer block (denoted by output$_1$ and output$_{\text{last}}$). We investigated three sequences with different lengths: 40, 240, and 1006.

As we can see in the top track (the non-rotated track), ChordMixer correctly identifies the relevant positions around the markers. The pattern is clearly visible for all sequences and in both output$_1$ and output$_{\text{last}}$). In output$_1$, high values basically reflect the rotation scales, while in output$_{\text{last}}$ the pattern spreads wider. Especially, output$_{\text{last}}$ demonstrates multi-scale attentions, where the upper tracks (i.e., those with no or small-scale rotations) show local attentions, while the lower tracks (i.e., with larger-scale rotations) show more even values.

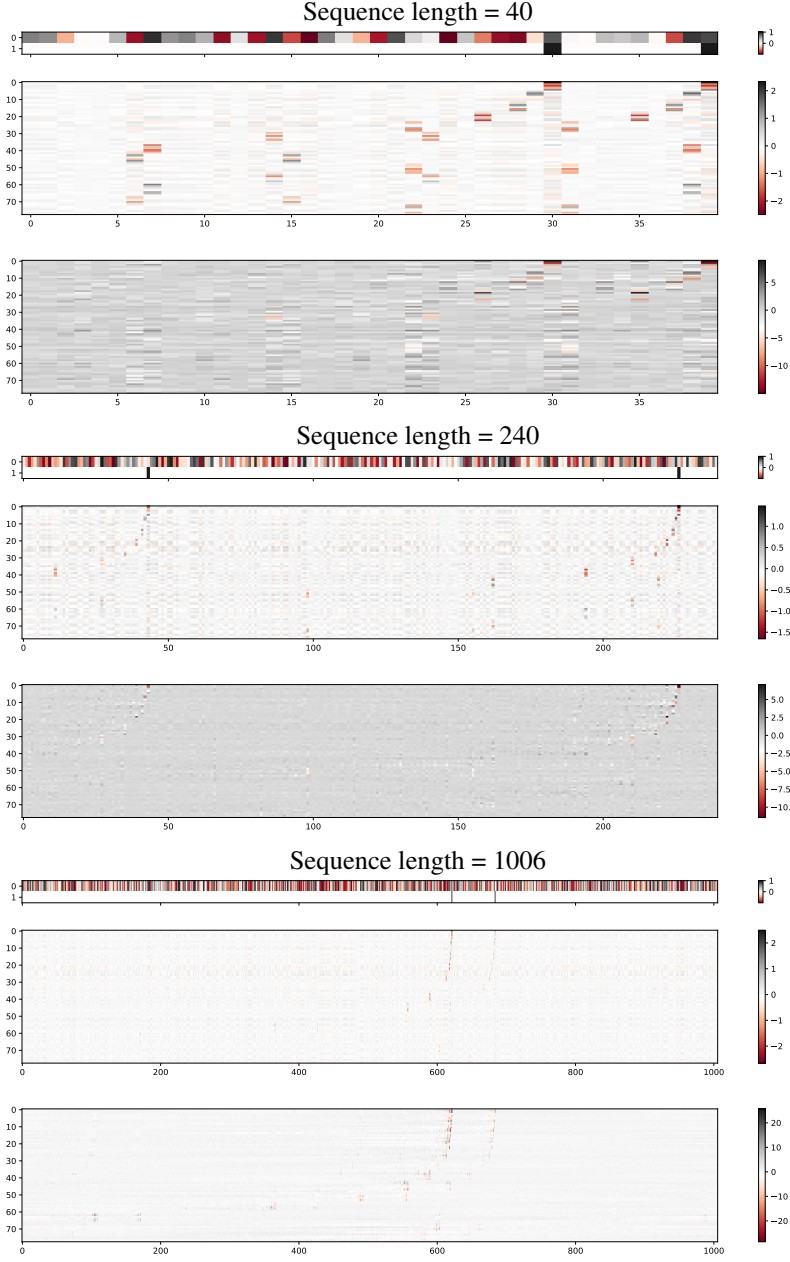

Figure 8: Visualizations of three examples in the Adding problem ($\lambda = 200$) with the ChordMixer model. The examples have different lengths. For each example, the upper chart corresponds to the input sequence, where the color map ranges from red (-1) to black (+1), with white correspond to 0; the middle and bottom charts are matrices after applying the first and the last ChordMixer blocks, respectively.

# E    MORE RESULTS FROM ABLATION STUDY

In the main paper Section 4.4, we have included ChordMixer, Poolformer, and Cosformer, with three pooling strategies CLS, AVG, and FLAT. Here we provide more extensive results, including four other models (Transformer, Reformer, Linformer, and Nyströmformer) and two other strategies (TRUNC and SEGM).

TRUNC means truncation. That is, if a sequence is longer than the truncation threshold $\psi$ (we used $\psi = 16k$), the part longer than $\psi$ is discarded. For sequences shorter than $\psi$, we padded them to length $\psi$. In TRUNC, the attention model output is flattened and followed by a linear predictor.

SEGM means segmentation. If a sequence is longer than the segmentation threshold $\beta$ (we used $\beta = 2000$), it is chunked into segments of length $\beta$. For sequences shorter than $\beta$, we padded them to length $\beta$. The compared methods using SEGM are first applied to all segments. Then the predictor outputs on individual segments are averaged for the final decision. Different from TRUNC, SEGM allows the compared method to access all elements in a sequence.

TRUNC and SEGM were used for those models that are not scalable for the whole longest sequence, where only pieces of the sequences are fed to the models. These strategies assume that the characterizing patterns appear only locally (within $\psi$ or $\beta$). However, they might lose information because they drop long interactions in the truncation or segmentation.

The more extensive ablation study results are shown in Figure 9. The conclusion in the main paper remains: our method is better than the other compared methods even for all tested strategies. ChordMixer achieves the best ROC-AUC in most sequence length percentile, especially for the long sequences. The additional compared methods and strategies do not lead to better results.

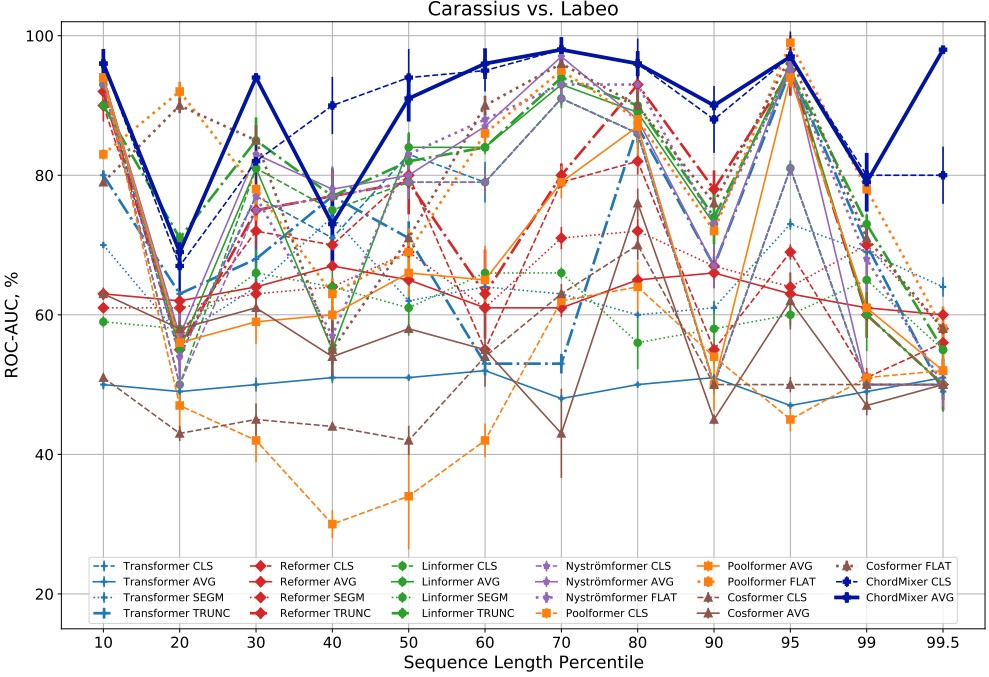

Figure 9: Ablation study on seven neural attention models with different pooling methods. Figure is better read in colors.

# F    PADDING STRATEGY

For the results reported in the main paper we used zero padding to the maximum sequence length across the whole data set (Pad to Overall Max). However, some of used methods allow using different sequence lengths during training. For such methods we zero-pad sequences to the maximum sequence

length in a batch (Pad to Batch Max) to further improve training and inference speed. We also empirically verified that there is no significant change in resulting performance when switching padding strategies (see 3).

Table 3: Accuracy comparison of different models with two pooling strategies on the Long Document Classification task. Pad to Overall Max and Pad to Batch Max denote padding to the maximum sequence length in the whole data set and in a batch, respectively. Reformer gets an Out-of-Memory error in both setups even when processing sequence by sequence.

| Model | Pad to Overall Max | Pad to Batch Max |
|---|---|---|
| LongFormer | 75.3% | 75.7% |
| Nyströmformer | 73.1% | 73.0% |
| Poolformer | 67.0% | 68.1% |
| Cosformer | 72.3% | 72.6% |
| Reformer | Out-of-Memory | Out-of-Memory |

## G  EMPIRICAL STUDY ON REACHING PROBABILITIES IN A CHORD NETWORK

At first glance, the Chord connection protocol looks asymmetric: in each hop, a node is connected to a few specific nodes in one direction. There is a question: if we perform a random walk from one node, how are its reaching probabilities to all nodes after several hops? It has been shown in the original Chord paper that any node can use $O(\log_2 N)$ hops to reach all $N$ nodes. Here we perform an empirical study on the distribution of the reaching probabilities.

We consider $N = 5000$ for example. We first normalize the adjacency matrix of the Chord graph $A$ to be right stochastic (i.e., rows sum to 1). Denote $\widetilde{A}$ the normalized matrix. Then $\widetilde{A}^{\lceil \log_2 N+1 \rceil} = \widetilde{A}^{14}$ gives the reaching probabilities from one node to another (including itself) after 14 hops. Because the circle is symmetric, we can pick any source node, e.g., the first, and plot the histogram of its reaching probabilities to all nodes as targets.

The result is shown in Figure 10. We can see that the reaching probabilities concentrate around the uniform probability $2 \times 10^{-4}$ with a small deviation (std=$3 \times 10^{-5}$), which justifies that in a ChordMixer network, each output position can receive information from all input positions with a nearly equal chance.

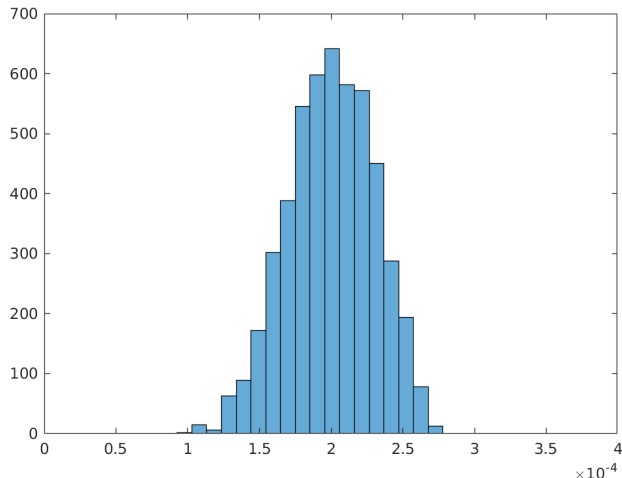

Figure 10: Histogram of reaching probabilities from a node to all nodes in a Chord network for $N = 5000$.

## H    PEAK MEMORY AND RUNNING TIME COMPARISON

Table 4: Comparison of peak memory (GB) and average running time (milliseconds) per sequence for the Adding problem. For all models, except for Linformer, we applied padding to the maximum sequence length within a batch. We have used a Linux machine with one Tesla-V100 GPU (32GB memory).

| Model | $\lambda = 200$ (longest=6.7k) | | $\lambda = 128k$ (longest=1.5M) | |
|---|---|---|---|---|
| | Peak Memory | Running Time | Peak Memory | Running Time |
| LSTM | 1.5 | 460 | - | - |
| Tree-LSTM | 25.3 | 9831 | - | - |
| 1D-CNN | 1.4 | 3.3 | - | - |
| TCN | 1.4 | 5.1 | - | - |
| Transformer | 1.9 | 9.8 | - | - |
| Linformer | 1.6 | 12.5 | - | - |
| Reformer | 2.0 | 23.3 | - | - |
| Longformer | 1.7 | 11.1 | - | - |
| Luna | 1.7 | 16.6 | - | - |
| S4 | 1.4 | 3.2 | - | - |
| Poolformer | 1.4 | 3.9 | 13.6 | 378 |
| Cosformer | 1.5 | 8.7 | 22.6 | 994 |
| ChordMixer | 1.5 | 4.9 | 23.1 | 604 |

## I    LONG RANGE ARENA PUBLIC BENCHMARK

We compared ChordMixer with Transformer and several of its variants on the Long Range Arena benchmark (Tay et al., 2020). Table 5 shows the accuracies of the compared methods on five LRA tasks. We can see that ChordMixer is substantially more accurate than the X-formers in all tasks. Similar to (Tay et al., 2020, Figure 3), we add ChordMixer to the plot of the relationship between accuracy and inference speed/memory footprint (see Figure 11). As we can see, ChordMixer consumes a similar amount of memory as the approximated variants of Transformer. Although Nyströmformer, Linformer, and Performer are faster, they sacrifice much prediction accuracy (more than 20% below ChordMixer).

Table 5: Classification accuracy of ChordMixer and X-formers on the LRA tasks. The performance of the competing models was taken from Tay et al. (2020) and Xiong et al. (2021). Boldface numbers are winners in each task, while the underlined are runner-ups.

| Model | ListOps $N = 2000$ | Text $N = 4000$ | Image $N = 1024$ | Retrieval $N = 4096$ | Pathfinder $N = 1024$ | PathfinderX $N = 16384$ |
|---|---|---|---|---|---|---|
| Transformer | 36.37 | 64.27 | 42.44 | 57.46 | 71.40 | ✗ |
| Longformer | 35.63 | 62.58 | 42.22 | 56.89 | 69.71 | ✗ |
| Linformer | 37.70 | 53.94 | 38.56 | 52.27 | 76.34 | ✗ |
| Reformer | 37.27 | 56.10 | 38.07 | 53.40 | 68.50 | ✗ |
| Performer | 18.01 | 65.40 | 42.77 | 53.82 | 77.05 | ✗ |
| Nyströmformer | 37.15 | 65.52 | 41.58 | 79.56 | 70.94 | ✗ |
| ChordMixer | **60.12** | **88.82** | **89.98** | **90.17** | **96.69** | **98.63** |

We do not include LRA in our main paper because it is not suitable for our problem setting: the sequences in each LRA task have the same length, and they are rather short compared to those in our experiments. See the statistics in Table 6.

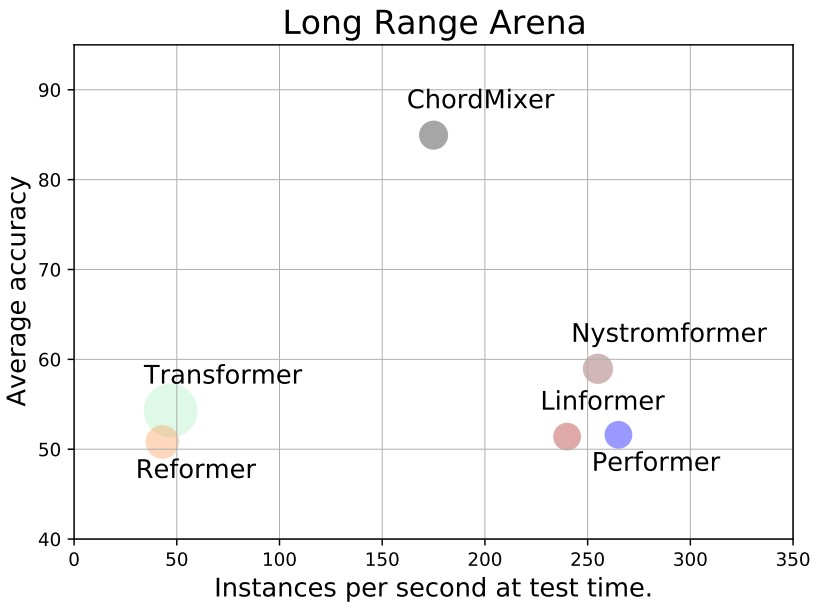

Figure 11: Accuracy (y-axis), speed (x-axis), and memory footprint (size of the circles) of different models measured on the LRA benchmark at test time.

| Task source | Task name | Sequence length |
|---|---|---|
| LRA | ListOps | 2K |
| LRA | Text | 4K |
| LRA | Retrieval | 4K |
| LRA | Image | 1K |
| LRA | PathFinder | 1K |
| LRA | PathFinder-X | 16K |
| Our experiments | Adding problem (base_length=128k) | min 32, median 134k, max 1.5M |
| Our experiments | Long document classification | min 4.5K, median 39K, max 131K |
| Our experiments | Carassius vs. Labeo | min 79, median 651, max 100K |
| Our experiments | Mus vs. Rattus | min 32, median 979, max 261K |
| Our experiments | Danio vs. Cyprinus | min 34, median 868, max 261K |
| Our experiments | Sus vs. Bos | min 63, median 664, max 447K |

Table 6: Length statistics of the data sets in our experiments and the Long Range Arena (LRA) data sets

## J SELF-SUPERVISED PRE-TRAINING WITH CHORDMIXER FOR GENETIC VARIANT CLASSIFICATION

In the main paper, we focused on ChordMixer for supervised learning tasks. We have been developing a framework where ChordMixer is used for self-supervised pre-training as well. Here we present some preliminary results for genetic variant importance prediction using DNA sequences.

An essential task in biology is to predict the influence of genetic variants on cell-type-specific gene expression, in order to inform fine-mapping of the many thousands of noncoding associations with phenotypes of interest from genome-wide association studies (GWAS). A genetic variant comprises a segment of gene sequence, a varied position, and the varied base at the position. The Genetic Variant Classification task is to predict whether a given variant effect is significant or not.

We have used the data set collected by Avsec et al. (2021). For self-supervised pre-training, we followed the Masked Language Modeling approach by using the non-masked bases (70%) to infer the masked bases (30%). In the pre-training, we have used over 100,000 randomly sampled human gene segments. The pre-training network employs the autoencoder architecture, where both encoder and decoder are ChordMixer networks. For supervised learning (fine-tuning), we extracted DNA sequences with up to 20k sequence length to construct the data set, which includes 97 992 instances over 49 tissues. In classification, we employ a light-weight classifier (average pooling + linear classifier) with the features output from the pre-trained encoder.

Table 7 shows the preliminary results, where we compared ChordMixer with the SOTA method in the field Enformer (Avsec et al., 2021). We can see that using ChordMixer as the classifier (without pre-training) is already substantially better than Enformer. The ROC-AUC is further improved by 3.7% by using the pre-trained encoder as a feature extractor.

Table 7: Preliminary results of comparing Enformer and ChordMixer on Genetic Variant Classification. The performance metric is the mean area under the receiver operating characteristic curve (ROC-AUC).

| Model | ROC-AUC$\times 100\%$ |
| --- | --- |
| Enformer (Avsec et al., 2021) | 70.5 |
| ChordMixer (without pre-training) | 84.9 |
| ChordMixer (with pre-training) | 88.1 |

