# OpenReview forum: "ChordMixer: A Scalable Neural Attention Model for Sequences with Different Length"
_ICLR.cc/2023/Conference — ICLR 2023 poster_

### Official Review · Reviewer_Bvdo · 2022-10-17

**Confidence:** 3
**Correctness:** 4
**Technical Novelty And Significance:** 3
**Empirical Novelty And Significance:** 3
**Recommendation:** 8

**Clarity, Quality, Novelty And Reproducibility:**

- The paper is clean and straigthforward to read.
- The main idea of the paper is very simple and intuitive to follow.
- The proposed approach is somehow novel. The idea of using Chrod in the context of sequence modelling has already been explored in Paramixer. Nevertheless, ChordMixer is simpler and allows for variable length sequence.
- The authors propose details about the model and the data used to train/eval it. Moreover, they provide the source code to train the model.

**Strength And Weaknesses:**

### Strengths
- The paper is well written and easy to follow. The authors did a good job on related works and provide a clear explanation of the proposed model.
- The problem tackled on this paper—modelling attention in very large sequences—is of extreme practical importance in many fields of machine learning.
- The proposed idea is very simple and intuitive. With a relatively simple and straightforward approach based on a P2P protocol, the authors are able to overcome some issues of current methods (eg, variable length and good performance in very long sequences).

### Weaknesses
- The paper only show results on very toysh tasks. Although I understand that even though standard models fail on those tasks, it would be much more interesting to see results in more realistic tasks (and/or on unsupervised pre-training).
- It is unclear to me why some models appears in the evaluation of some tasks, but not others. Eg, S4, Luna appear on the “Adding problem” but not on others and the “long document classification” only has 4 models. ow does the omitted models perform on those tasks? are they comparable or better than ChordMixer?
- As pointed by the authors, this approach also uses the Chord protocol. However, Paramixer does not show on any experimental comparison. It would be nice to compare ChordMixer and Paramixer model, in particular what are the pros and cons of each architecture.
- As mentioned by the authors, training requires that every batch contains sequences of the same length. Is this an issue? Can this bias the model in some particular way?

**Summary Of The Paper:**

In this paper, the authors propose ChordMixer: a simple neural network building block that is able to mix tokens within a sequence. Each ChordMixer block consists of a rotation layer followed by 2-layer MLP. Compared to previous methods, ChordMixer allows attention over variable input length and performs favourably on very long sequence tasks. The authors show results on three toy tasks: synthetic addition problem, long document classification and DNA sequence-based taxonomy.

======== Post-rebuttal update ========

After reading other reviewers' comments and the authors' rebuttal, I decided to upgrade my score to "8: accept, good paper". The model is simple and clever and the experimental results are convincing. Hopefully the code will be available. Congratulations!

**Summary Of The Review:**

Based on the comments above, I am rating this paper as "marginally above acceptance threshold". The mode is simple, somewhat novel and they show better results than current models on toy datasets with very long sequence lengths. It would be nice if the authors could clarify the points I mentioned on the "weaknesses" section.

---

> ### Author Response · Authors · 2022-11-17
> **Thanks for your review. Please find our answers below.**
>
> **Q:** The paper only shows results on very toysh tasks. Although I understand that even though standard models fail on those tasks, it would be much more interesting to see results in more realistic tasks (and/or on unsupervised pre-training).
>
> **A:** No, our tested tasks are not toysh. The sequences are very long with high variation. Supervised inference over such sequences was unsolved before. Both long document classification and DNA sequence-based taxonomy classification are real-world applications.
>
> Our method also works for self-supervised pre-training. We have added a preliminary result in the appendix (Section J), where pre-training with our network architecture improves the prediction accuracy for genetic variants.
>
>
> **Q:** It is unclear to me why some models appears in the evaluation of some tasks, but not others. Eg, S4, Luna appear on the “Adding problem” but not on others and the “long document classification” only has 4 models. How does the omitted models perform on those tasks? are they comparable or better than ChordMixer?
>
> **A:** The reason is given at the end of Section 4.2: these methods ran out of memory, and we had to drop them from the comparison.
>
> **Q:** As pointed by the authors, this approach also uses the Chord protocol. However, Paramixer does not show on any experimental comparison. It would be nice to compare ChordMixer and Paramixer model, in particular what are the pros and cons of each architecture.
>
> **A:** It is because Paramixer assumes that the input lengths are the same (first paragraph in Section 3.4). If we use padding to ensure all sequences have the maximum length, Paramixer will crash with an out-of-memory error.
>
> **Q:** As mentioned by the authors, training requires that every batch contains sequences of the same length. Is this an issue? Can this bias the model in some particular way?
>
> **A:** No, we do not require that every batch contains sequences of the same length. Instead, we require that the sequences in a batch have the same $\lceil\log_2 N\rceil$ value (last paragraph in Page 3). For example, four sequences with lengths 5000, 6000, 7000, and 8000, respectively, can be in a batch because their $\lceil\log_2 N\rceil$ values are all 13. This requirement brings little bias to the model.

---

### Official Review · Reviewer_pZas · 2022-10-25

**Confidence:** 4
**Correctness:** 2
**Technical Novelty And Significance:** 2
**Empirical Novelty And Significance:** 2
**Recommendation:** 5

**Clarity, Quality, Novelty And Reproducibility:**

The math in the paper isn't clear. For example, symbols "x" and "z" are used without definition or description in eq 1. The meaning of the indices i, j, and k also needs to be inferred from the context, which makes it hard to understand the work.

The novelty of this paper is limited as some pattern-based methods are pretty similar to this work.

I noticed that the codes of this paper have been released, along with all the data used in the paper, thus I do not doubt the reproducibility of ChordMixer.

**Strength And Weaknesses:**

Strengths:

1. ChordMixer is theoretically guaranteed to have sublinear time and space complexity, which makes it efficient on long sequence tasks.
2. The proposed method is simple and can be efficiently implemented with modern hardware.
3. ChordMixer is shown to be effective on the tasks presented in the paper.

Weaknesses:

1. Experiments should be conducted on more standard datasets, e.g. Long-Range Arena. Given that there are tons of work on long sequence transformers, comparing them one by one is unrealistic, thus testing ChordMixer on such a dataset is a preferable way. I would expect the ChordMixer to be a point in fig 3 of Tay et al. (2021) (the LRA paper). Working with a new dataset makes it hard to judge the performance of ChordMixer.
2. In section 4.2, I don't know why other models all need zero padding to align with the maximum length. Some models, e.g. Reformer and Linformer do not have any constraints on the input length and do not need the inputs to be of equal lengths.
3. ChordMixer might be viewed as a pattern-based method in the taxonomy of Tay et al. (2020), then some baselines seem necessary. E.g. random pattern can similarly achieve a long receptive field with a high probability, and longformer (sliding window + dilated pattern + global token) can also achieve a large receptive field for relatively long sequences, although I think a larger receptive field does not necessarily lead to better interactions between distant tokens.
4. The presentation needs to be improved. I would give more details in the next section.

Questions:

1. In section 2, I don't think Longformer and ETC use learnable side memory modules (or do you mean the global token?).
2. ChordMixer is designed to have sparse routes, while some sequences, such as natural language, and adjacent tokens could be very important contexts. Do you think ChordMixer may be less efficient to capture this information?

**Summary Of The Paper:**

The paper introduces a method to extend neural attention to longer text with limited computations. With inspiration from the P2P Chord protocol, it rotates the token indices to allow the model to attend to tokens in different scales of distances. It is proved to have a large receptive field compared to traditional transformers while preserving sublinear space and time complexity. Experiments are conducted on both synthetic arithmetic problems, long document classification, and DNA taxonomy tasks. The proposed method, named ChordMixer, is shown to be more effective than its baseline models.

**Summary Of The Review:**

The paper proposes a method that mimics the Chord protocol to allow distant attention for long sequence encoding. Although shown to be effective on newly proposed data, more popular datasets should be used so the results can be more convincing. Also, the novelty of this paper is limited given many pattern-based methods have already been proposed with similar ideas. I would suggest a rejection for this paper.

---

> ### Author Response · Authors · 2022-11-17
> **Thanks for your review. Please find our answers below.**
>
> **Q:** Experiments should be conducted on more standard datasets, e.g. Long-Range Arena. Given that there are tons of work on long sequence transformers, comparing them one by one is unrealistic, thus testing ChordMixer on such a dataset is a preferable way. I would expect the ChordMixer to be a point in fig 3 of Tay et al. (2021) (the LRA paper). Working with a new dataset makes it hard to judge the performance of ChordMixer.
>
> **A:** We have added the comparison results on LRA (see Appendix I), where ChordMixer performs well in terms of high accuracy and reasonably low cost in time and memory.
>
> We do not include LRA in our main paper because it is not suitable for our problem setting: the sequences in each LRA task have the same length, and they are rather short compared to those in our experiments. See the following statistics:
>
> | Task source | Task name  | Sequence length |
> | ----------- | ---------- | --------------- |
> | LRA | ListOps | 2K |
> | LRA | Text | 4K |
> | LRA | Retrieval | 4K |
> | LRA | Image | 1K |
> | LRA | PathFinder | 1K |
> | LRA | PathFinder-X | 16K |
> | Our experiments | Adding problem (base_length=128k) | min 32, median 134k, max 1.5M |
> | Our experiments | Long document classification | min 4.5K, median 39K, max 131K |
> | Our experiments | Carassius vs. Labeo | min 79, median 651, max 100K |
> | Our experiments | Mus vs. Rattus | min 32, median 979, max 261K |
> | Our experiments | Danio vs. Cyprinus | min 34, median 868, max 261K |
> | Our experiments | Sus vs. Bos | min 63, median 664, max 447K |
>
> You can see that the experiment tasks in our main paper are more challenging, and the wins of ChordMixer are convincing.
>
> **Q:** In section 4.2, I don't know why other models all need zero padding to align with the maximum length. Some models, e.g. Reformer and Linformer do not have any constraints on the input length and do not need the inputs to be of equal lengths.
>
> **A:** We have added results Transformer, Longformer, Reformer, CosFormer, and Nystromformer, where they apply zero-padding to align with the longest sequence in a batch. See Section F in the appendix. Although the within-batch padding strategy improves speed for the listed methods, it does not affect either their accuracy or peak memory (at least one batch is still padded to the overall longest). Therefore the conclusion of our wins still holds.
>
> The current implementation of Linformer requires a fixed sequence length in their network initialization. Therefore, we had to set the fixed length to the maximum sequence length.
>
>
> **Q:** ChordMixer might be viewed as a pattern-based method in the taxonomy of Tay et al. (2020), then some baselines seem necessary. E.g. random pattern can similarly achieve a long receptive field with a high probability, and longformer (sliding window + dilated pattern + global token) can also achieve a large receptive field for relatively long sequences, although I think a larger receptive field does not necessarily lead to better interactions between distant tokens.
>
> **A:** We have added Longformer to the comparison for the classification tasks (see the revision, Section 4.2 and Section 4.3, Appendix I), where ChordMixer clearly wins over it.
>
> Large receptive fields are necessary. For example, it is known that long-range interactions between DNA elements are common and can be up to 20,000 bases away (Gasperini et al., 2020). It is impossible to model such long-range interactions without large receptive fields.
>
>
> **Q:** In section 2, I don't think Longformer and ETC use learnable side memory modules (or do you mean the global token?).
>
> **A:** Yes, we meant global memory tokens. See "Efficient Transformers: A Survey" by Yi Tay, Mostafa Dehghani, Dara Bahri, Donald Metzler, accepted to ACM Computing Surveys 2022 (Section 3.1, bullet Neural Memory).
>
> **Q:** ChordMixer is designed to have sparse routes, while some sequences, such as natural language, and adjacent tokens could be very important contexts. Do you think ChordMixer may be less efficient to capture this information?
>
> **A:** ChordMixer does not miss local signals. Appendix G shows that the learning signal can reach all sequence positions with nearly uniform probabilities. Section 4.2 shows that ChordMixer works well for text documents.
>
> Over-emphasis on locality could ease training for some small data sets. However, it hinders the discovery of long-range interactions. For big data, it is more objective to use a uniform prior and let the data decide the attention range.

---

> > ### Author Response · Authors · 2022-11-17
> > **Part 2.**
> >
> > **Q:** The math in the paper isn't clear. For example, symbols "x" and "z" are used without definition or description in eq 1. The meaning of the indices i, j, and k also needs to be inferred from the context, which makes it hard to understand the work.
> >
> > **A:** Please notice the symbol $x$ is defined in first paragraph of Section 2. We have recapitulated the tensor dimensions in Section 3.2 of the revision.
> >
> > **Q:** The novelty of this paper is limited as some pattern-based methods are pretty similar to this work.
> >
> > **A:** No, your comment is not true.
> >   - ChordMixer is different from sparse variants of Transformers because we do not use scaled dot-products.
> >   - ChordMixer uses a sparse structure that does not exist in the literature of sparse Transformers.
> >   - ChordMixer clearly wins in all presented tasks as well as in the LRA benchmark over the existing sparse variants of Transformers.

---

> ### Comment · Reviewer_pZas · 2022-12-03
> **Response to the rebuttal**
>
> Thanks for the long reply. I decided to raise my score because of the addition of the public benchmark. It makes sense that ChordMixer aims to solve problems with super long sequences, while LRA are relatively short.

---

### Official Review · Reviewer_9e1K · 2022-10-26

**Confidence:** 4
**Correctness:** 4
**Technical Novelty And Significance:** 3
**Empirical Novelty And Significance:** 3
**Recommendation:** 8

**Clarity, Quality, Novelty And Reproducibility:**

 The paper is well-written and easy to follow. The idea of using the Chord protocol to enable long-range interactions is novel and also theoretically well-motivated. The experiments compare against recent attention as well as LSTM and CNN to showcase the efficacy. Code and details are provided for reproducibility.


**Strength And Weaknesses:**

**Strengths**

- The paper is clearly written. The design of the ChordMixer block is novel, conceptually simple, and well-motivated from the P2P network. The figures on the Chord protocol and the rotate layer make it easy to understand.
- The experimental results on the artificial and real-world sequences showcase that ChordMixer outperforms recent efficient attention models when dealing with long sequences. The code in the supplementary and the details should make the work easily reproducible.


**Questions and Weaknesses**:

- Page 1 paragraph 2: "Although numerous architectures such as Transformer and its variants have been proposed, they usually assume constant input length." The assumption/requirement stems from efficient GPU processing, thus, it would be better to rephrase this as "For efficient batch processing, architectures such as...."

- The paper mentions "We believe after the mixing, elements at every position summarize good information from all input positions.". While at the end of $log(N)$ steps, each output token receives information from each input token, the mixing doesn't seem uniform. For instance, consider the case of $i$-th token. Tokens $i+1$ and $i+2$ would mix with it in a single step. In the next hop/block $i+1$ token would again mix with $i+2$ token carrying information received from $i$-th token. In contrast, $\exists$ a token that mixes with the $i$-th token only at the last hop. Could it affect the performance or introduce inductive biases? It would be good to discuss this case. Note that the paper covers reaching probabilities in Appendix, however, I am not sure if that answers the question completely.

- The paper says "A ChordMixer network consists of $log(N_{\text{max}})$ blocks, where $N_{\text{max}}$ is the length of the longest sequence." Does each input get processed with the same number of blocks or do you use fewer blocks for shorter sequences? Another question is what happens if you obtain a sequence of length $2 \times N_{\text{max}}$ during inference? Though this may not be likely for datasets with long sequences.

**Summary Of The Paper:**

This paper proposes a new neural attention model called "ChordMixer" to enable long-range interactions for sequences with variable lengths. The proposal takes inspiration from the Chord protocol in P2P networks. The ChordMixer consists of a series of ChordMixer blocks where in each block, an input token with $d/M$ dimensions (M = $log(N) + 1$) first interacts with $k=log(N)$ peers/tokens via the rotation layer. The rotation layer concatenates the representation from peers to its own to get a $d$ dimensional representation for the token. Next, each token is processed by a mixer layer comprising a two-layer MLP which enables sharing of information. A series of $log(N)$ blocks ensure a full-receptive field, i.e., all output tokens can be influenced by each input token.

Experiments on tasks involving synthetic and real data show that ChordMixer is scalable to very long sequences while outperforming RNNs, CNN, and other efficient attention methods.

**Summary Of The Review:**

As explained above, the idea of ChordMixer is novel and the experiments are thorough. Thus I recommend acceptance for this work.

---

> ### Author Response · Authors · 2022-11-17
> **Thanks for your review. Please find our answers below.**
>
> **Q:** Page 1 paragraph 2: "Although numerous architectures such as Transformer and its variants have been proposed, they usually assume constant input length." The assumption/requirement stems from efficient GPU processing, thus, it would be better to rephrase this as "For efficient batch processing, architectures such as...."
>
> **A:** Fixed. Thanks.
>
> **Q:** The paper mentions "We believe after the mixing, elements at every position summarize good information from all input positions.". While at the end of $\log_2 N$ steps, each output token receives information from each input token, the mixing doesn't seem uniform. For instance, consider the case of $i$-th token. Tokens $i+1$ and $i+2$ would mix with it in a single step. In the next hop/block $i+1$ token would again mix with $i+2$ token carrying information received from $i$-th token. In contrast, $\exists$ a token that mixes with the $i$-th token only at the last hop. Could it affect the performance or introduce inductive biases? It would be good to discuss this case. Note that the paper covers reaching probabilities in Appendix, however, I am not sure if that answers the question completely.
>
> **A:** Yes, Appendix G exactly answers the concern. It shows that the probabilities of different target tokens receiving information from any source token are nearly the same after $\log_2 N$ blocks. There is little asymmetry.
>
> **Q:** The paper says "A ChordMixer network consists of $log(N_\text{max})$ blocks, where $N_\text{max}$ is the length of the longest sequence." Does each input get processed with the same number of blocks or do you use fewer blocks for shorter sequences? Another question is what happens if you obtain a sequence of length $2\times N_\text{max}$ during inference? Though this may not be likely for datasets with long sequences.
>
> **A:** We wrote that "$N_\text{max}$ is the length of the longest sequence or the longest range of possible interactions" (Section 3.2). So $N_\text{max}$ does not scope the training set only. If it is set to the longest range of possible interactions, we can truncate any too-long sequences to $N_\text{max}$ without losing interaction information.

---

### Decision · Program_Chairs · 2023-01-20

**Decision:**

Accept: poster

**Justification For Why Not Higher Score:**

N/A

**Justification For Why Not Lower Score:**

N/A

**Metareview: Summary, Strengths And Weaknesses:**

The paper proposes "ChordMixer", a new attention module for long-range modeling. Each block consists of a position-wise rotation layer and an elementwise MLP that is used to mix the input signals. There is a consensus in the reviewers that this is a solid paper and a good contribution to the current toolbox of sequence modeling. The main concern is around the experiments (where maybe more standard benchmarks such as LRA should be used). I agree with that but think that is not a big deal comparing to the contribution of the paper. I therefore recommend acceptance of this paper.

**Note From Pc:**

if the above contains the word "oral" or "spotlight" please see: "oral" presentation means -> notable-top-5% and "spotlight" means -> notable-top-25%. As stated in our emails, we are disassociating presentation type from AC recommendations